# Exploring interactions between socioeconomic context and natural hazards on human population displacement

Michele Ronco [1,3] ✉, José María Tárraga [1,3], Jordi Muñoz [1], María Piles [1], Eva Sevillano Marco [1], Qiang Wang [1], Maria Teresa Miranda Espinosa [2], Sylvain Ponserre [2] & Gustau Camps-Valls [1]

Climate change is leading to more extreme weather hazards, forcing human populations to be displaced. We employ explainable machine learning techniques to model and understand internal displacement flows and patterns from observational data alone. For this purpose, a large, harmonized, global database of disaster-induced movements in the presence of floods, storms, and landslides during 2016–2021 is presented. We account for environmental, societal, and economic factors to predict the number of displaced persons per event in the affected regions. Here we show that displacements can be primarily attributed to the combination of poor household conditions and intense precipitation, as revealed through the interpretation of the trained models using both Shapley values and causality-based methods. We hence provide empirical evidence that differential or uneven vulnerability exists and provide a means for its quantification, which could help advance evidence-based mitigation and adaptation planning efforts.

Throughout history, climate has played a role in influencing human mobility, with populations often resorting to movements as an adaptive response to environmental changes, seeking to enhance their prospects for survival[1]. Notably, current observations suggest that recent changes in climate may also be influenced by anthropogenic factors, which have the potential to disrupt traditional lifestyles[2–5]. The rapidly evolving climate, characterized by an increasing frequency and severity of extreme weather events, may pose challenges to the effective implementation of mitigation and adaptation measures[6–10]. In light of these factors, the likelihood of an increased incidence of displacement as a response to these adversities deserves consideration[11,12]. Disaster displacement involves situations where individuals are forced to leave their usual residential areas due to a natural or anthropogenic hazard[13]. Yet wealth and resources are not equally distributed, and the population is concentrated in low and middle-income countries[14–16]. This could exacerbate existing challenges in these regions, making it harder for them to cope with the effects of environmental hazards and climate stressors[17–21].

It is undeniable that the relation between weather and population movements is complex[22–27]. Despite the widespread quest for one main trigger, human mobility has a multi-causal nature[28]. There is never a single reason why people move but rather an intricate tangle of heterogeneous and interacting factors[4,29,30]. Further complexity is added by the confounding role that natural hazards play in damaging local livelihoods, economic activities, and infrastructures. Also, the relationship between humans and the environment is mediated by how people perceive their environmental context, including subjective factors that may pose challenges when incorporating them into statistical models[31–36]. Ultimately, increased insecurity, such as armed conflict, food and water scarcity, and other life-threatening conditions, can lead to forcibly displaced people. Casting disaster risk as the intersection between hazard, exposure, and vulnerability can be particularly useful to examine the linkage between displacements and environmental stress[10,29,37–41]. In this context, vulnerability refers to conditions that can increase a community's likelihood of experiencing adverse effects from natural or human-induced hazards, including

[1]Image Processing Laboratory (IPL), Universitat de València, Valencia, Spain. [2]Internal Displacement Monitoring Centre (IDMC), Geneva, Switzerland. [3]These authors contributed equally: Michele Ronco, José María Tárraga. ✉e-mail: michele.ronco@uv.es

physical, social, and economic factors, such as poverty or inadequate infrastructure. Exposure pertains to human assets in hazard-prone areas, such as people, structures, cropland, homes, and manufacturing capacity, which could be affected by disasters. The more severe a weather event, the greater its impact could be on human displacement, provided that vulnerable people and livelihoods are exposed in the affected area. Then, whether there will be weather-induced displacements in response to a hazard and, if so, the number of people moving will depend crucially on the economic resources and the adaptive capacity of the impacted community[42–44].

Past research has rarely considered together all these three dimensions of the problem, and analytic models have often assumed simple linear relationships (with some notable exceptions[45–52]), often neglecting the intrinsic non-linearity of the problem[53]. It is worth noticing that many research studies focus on international migration[54,55]. However, weather hazards most likely generate internal displacements, i.e. short-distance movements typically from rural to urban areas within the borders of a country[53,56–59]. Depending on the definition of the target variable of interest, the results point toward moderate or no evidence for environmental factors as human mobility drivers[55]. Yet another major limitation is represented by the lack of data in terms of availability, completeness, and reliability[60]. Collecting reliable data on people's movements is notoriously difficult. Only in the last years, more systematic monitoring programs at a large scale have been launched[61]. Among other factors, the results of the study depend on the selected countries, the type of mobility in question, the period considered, and the chosen predictors. What are the most relevant data to analyze the problem remains an open debate.

Here, we study human internal displacement induced by sudden-onset hazards at a global scale with data-driven machine learning (ML) algorithms. We employ ensemble models, specifically random forests (RFs)[62] and gradient boosting machines (GBMs)[63], to predict the number of new displacements of people (NDP)[61] registered in

concomitance with each hazard (flood, storm, or landslide) in the years 2016−2021. NDP refers to the estimated number of individuals who have been internally displaced from their habitual places of residence during a specific time period. We compare the performance of these models with a baseline linear model. Our prediction models utilize a diverse set of socioeconomic and environmental drivers on a national and disaster-specific scale (see details in Material and methods, cf. Fig. 1, and Supple. Information). The proposed approach avoids strong assumptions of variable relations or relevance and solely relies on observational data. In addition, being based on explainable AI (XAI)[64] and methods for causal effect estimation[65,66] (see details in Material and methods and Supple. Information), our analysis sheds light on the complex interactions between the involved and often mediating processes and drivers of people movements.

## Results

### The multivariate disaster-driven displacement problem

Understanding hazard-induced internal displacement is a multivariate complex problem. To alleviate the sufficiency assumption, asserting that all relevant variables influencing the target have been included, we collected a set of potentially explanatory covariates of different types (economic, weather, land specific) and granularity levels (polygon or national scale) for each disaster event (see Fig. 2). The hazard component is represented by precipitation[67] and wind speed (WS)[67]; exposure is given by nonlinear normalized difference vegetation index (kNDVI)[68], the fraction of agricultural land (%AgriLand)[69], elevation[70], affected area of the polygon (accounting for both exposed assets and people)[71] and population as a measure of human exposure[72]; finally, vulnerability is characterized by education expenditures (%EduExp)[69], Absolute Wealth Index (AWI)[73], global human modification (gHM) index as a measure of the anthropogenic action on land[74], and fatalities resulting from conflicts[75] (see details in Material and methods, cf. Table 1, and Supple. information). We select all those countries for

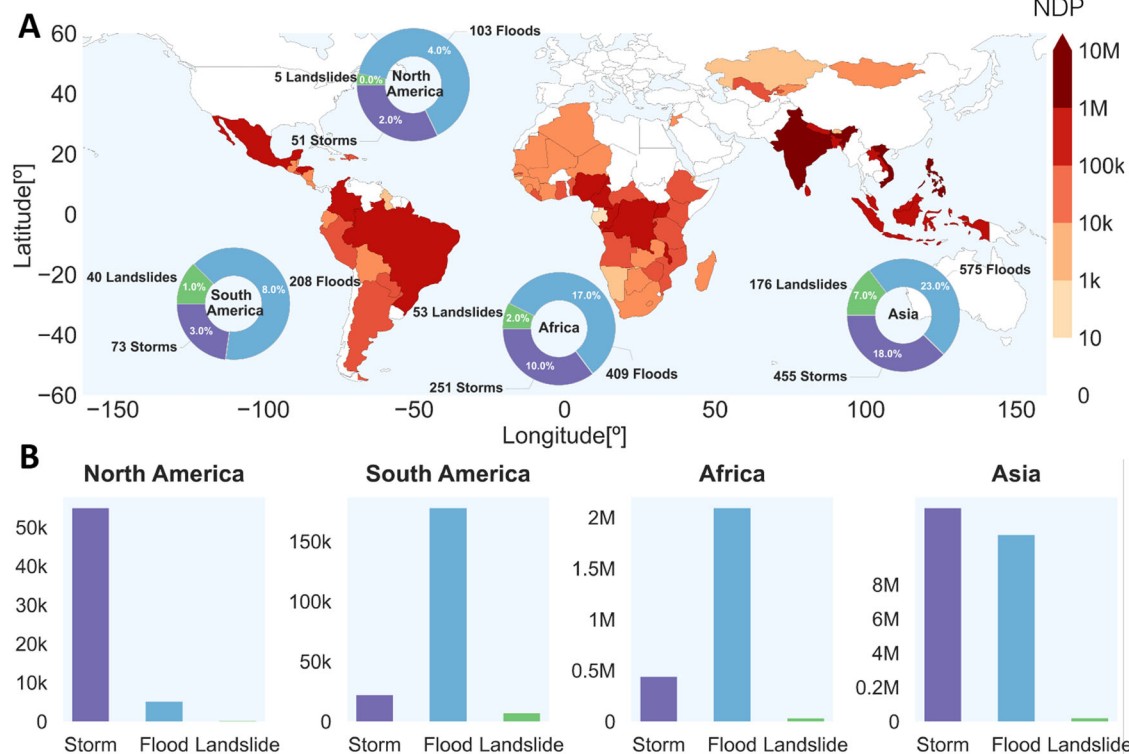

**Fig. 1 | Spatial distribution of newly displaced people (NDP) per sudden-onset disaster over selected countries for years 2016–2021. A** Colors represent the sum of NDP per country registered in the years under consideration; pie charts indicate the event counts and percentages with respect to the global number of events. **B** The total number of NDP per continent and hazard type occurred in the period of interest.

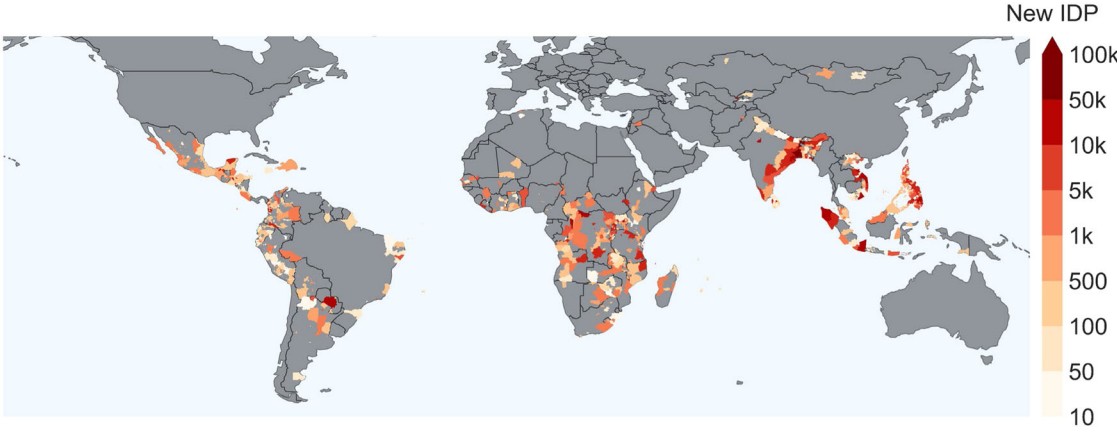

**Fig. 2 | Map of the areas impacted by sudden-onset hazards.** Polygons are given at the administrative level 1 or 2 or by a combination of the two, depending on the disaster-affected area. Color represents the total sum of newly displaced people (NDP) produced by hazards in each polygon in years 2016–2021.

## Table 1 | Features used to predict NDP

| Variable | Temporal aggregation | Spatial aggregation | Granularity | Source |
|---|---|---|---|---|
| 1. AWI | Max | Max | Polygon | Meta Data4Good[73] |
| 2. Precipitation | Max | Sum | Polygon | ERA5-Land (GEE)[67] |
| 3. 10m Wind Speed | Max | Max | Polygon | ERA5-Land (GEE)[67] |
| 4. kNDVI | Mean | Mean | Polygon | MODIS TERRA (GEE)[105] |
| 5. Population | Mean | Mean | Polygon | GPWv4[72] |
| 6. gHM | Mean | Mean | Polygon | CSP[74] |
| 7. Elevation | Mean | Mean | Polygon | NASA/CGIAR (GEE)[70] |
| 8. Conflict fatalities | Sum | Sum | Polygon | ACLED[75] |
| 9. Area | – | – | Polygon | OpenStreetMap[71] |
| 10. Education expenditures | – | – | National | SDG API[69] |
| 11. % Agricultural Land | – | – | National | SDG API[69] |

which the above data are available (see, in particular, AWI specifics[73]), avoiding gap filling and imputation of missing values. Due to these constraints and availability limitations, our focus is exclusively on low and middle-income countries as defined by the Demographic and Health Surveys Program[73], which are widely recognized as being particularly susceptible to the impacts of climate change[14–20,54].

To detect linkages between human mobility and weather hazards, a variety of indicators of population movements have been suggested, but their suitability has been challenged[76]. Here, we focus on NDP registered by the Internal Displacement Monitoring Centre (IDMC)[61] concomitantly with three forms of sudden-onset disasters, namely storms, floods, or landslides. IDMC database is the only aggregator of internal displacement data, with global coverage by type of disaster hazard and a consistent data model[77]. Displacement data are collected from January to December of each year. It is worth noting that the figures may include individuals who have experienced displacement more than once. Our dataset contains a total of 2400 disaster events in the period 2016–2021 (see details in Material and methods, and Fig. 1).

Aggregation of NDP at a continent level shows that Asia is by far the most impacted continent regardless of the type of disaster, see Fig. 1. This is particularly evident in the most densely populated coastal regions, often affected by severe storms. North America is also affected mainly by storms, but the impact of total movements of people is about a factor of $10^2$ lower compared to Asia. The other continents have more NDP associated with floods. Landslides represent a marginal component of displacements in all continents with respect to floods and storms.

Here, the question is about what conditions and combinations of drivers give rise to higher NDP and if these reveal to some extent the presence of differential vulnerability, a concept which already appeared in the literature[53,78–81]. To answer these questions, we adopt a rather agnostic data-driven modeling approach based on combining ML models with XAI and causality techniques.

**Modeling and understanding displacements with explainable AI**
We trained RFs and GBMs to estimate the logarithm of NDP using the set of covariates listed in Table 1, and then compared their performance to a linear regression (LR) baseline model. Models were extensively cross-validated, and test data bootstrapped to robustly estimate the performances (see Material and methods). Ensemble models achieved better goodness-of-fit $R^2$ and accuracy RMSE values in comparison with the LR baseline (see results in Fig. 3 and Table 2). Our results are comparable to results reported in similar analyses elsewhere[50,53,55], albeit direct comparability is not possible since different targets and data are used. Moreover, we observed a quantitative and measurable effect of weather variables on the model predictability (note the degradation of performance scores in Table 2 and the shift of the $R^2$ distribution in Fig. 3B). To estimate the statistical significance of this drop in the mean $R^2$, we counted how many times the hold-out $R^2$ of the RF without weather variables is equal or better than that of the RF trained with all covariates and obtained a $p$-value of 0.05. This supports the relevance of the predictors accounting for weather conditions.

The previous metrics tell us about the overall prediction performance only. Still, we are interested in understanding how the

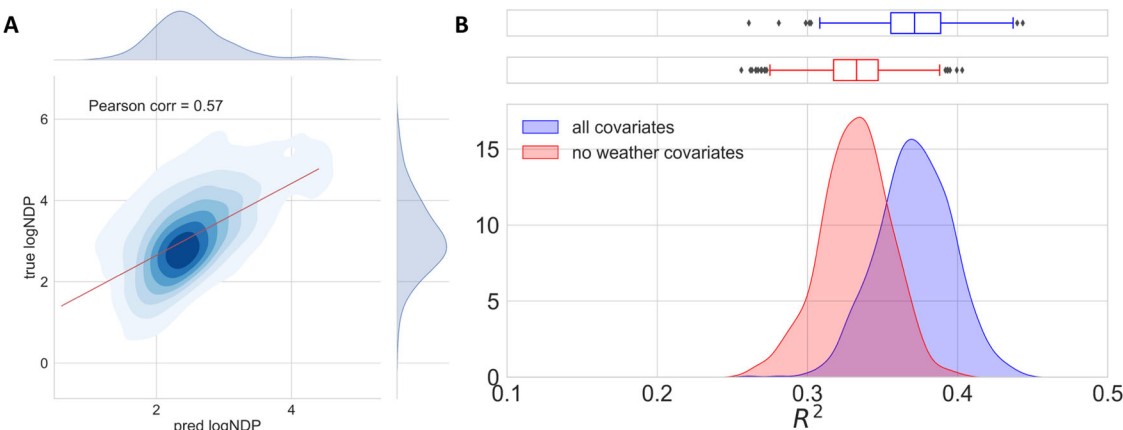

**Fig. 3 | Performance of the trained Random Forest models. A** Predictions versus true values in logarithmic scale were obtained by averaging over all test batches in the bootstrapping. The color levels show the density of points. The Pearson correlation is 0.57. **B** Distribution of the $R^2$ on the hold-out set for all the bootstrapping iterations. The blue density is obtained with all the covariates, while the red one is obtained by excluding the two weather variables, namely maximum precipitation accumulation and maximum 10 m wind speed.

**Table 2 | Performance of the models – coefficient of determination R², root mean square error (RMSE) and mean error (ME)– on the test (hold out) set obtained with bootstrapping using Linear Regression (LR), Random Forest (RF), and Gradient Boosting Machine (GBM)**

| Metric | LR (all) | GBM (all) | RF (all) | LR (no weather) | GBM (no weather) | RF (no weather) |
|---|---|---|---|---|---|---|
| 1. R² | 0.19 ± 0.02 | 0.36 ± 0.02 | 0.37 ± 0.02 | 0.16 ± 0.02 | 0.32 ± 0.02 | 0.33 ± 0.02 |
| 2. RMSE | 1.02 ± 0.02 | 0.91 ± 0.02 | 0.90 ± 0.02 | 1.04 ± 0.01 | 0.93 ± 0.02 | 0.93 ± 0.02 |
| 3. ME | −0.001 ± 1.0 | −0.003 ± 0.91 | −0.006 ± 0.90 | −0.001 ± 1.02 | −0.002 ± 0.93 | −0.005 ± 0.93 |

Results obtained with all covariates are compared with those found when excluding extreme weather factors represented by precipitation and wind speed. Average performances and their standard deviation have been calculated by using bootstrapping.

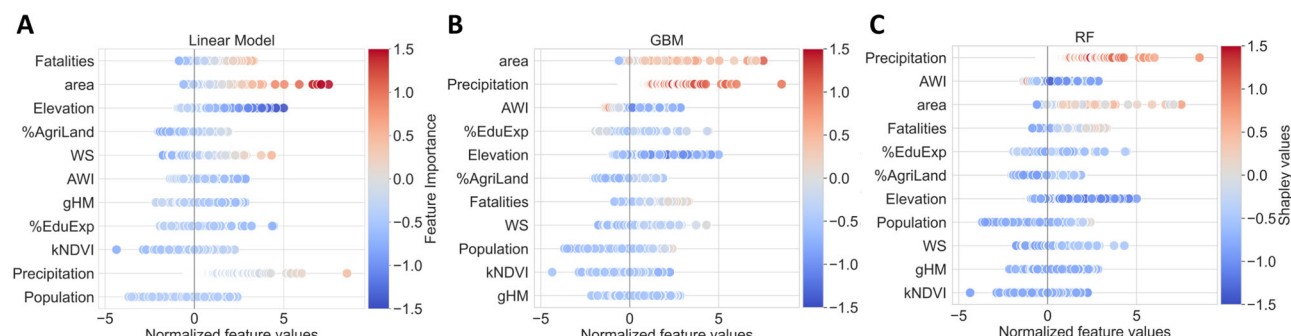

**Fig. 4 | Relation between the input features and their importance scores averaged over the different test batch configurations in the bootstrapping.** The horizontal axis represents the normalized feature values. At the same time, the color scale is given by the mean product between the weights of the linear model (**A**) and the feature value or the mean Shapley value per event for the Gradient Boosting Machines (**B**) and Random Forests (**C**). The covariates are displayed in decreasing order of importance.

model utilizes different factors to predict the NDP per event and separate the impact of various displacement drivers. The field of XAI helps get insight from non-parametric ML models[82]. Figure 4 reports the Shapley values[83–86] for the GBM (B) and the RF (C), a popular metric of XAI (see Methods section and Supple. information), to further understand the model mechanisms and the most relevant covariates. A positive Shapley value for a predictor means that such a predictor raises the value of the target, while negative values tend to lower it. Instead, the feature importance for the LR (A) is calculated simply by multiplying the weights by the corresponding predictor values for each instance.

Looking at the ranking in Fig. 4, we first notice that, overall, in agreement with previous studies[53], the vulnerability (e.g. AWI) and hazard (e.g. precipitation) variables are the most important, followed by exposure factors (e.g. area). This confirms that socioeconomic conditions play a crucial role in the magnitude of displacements. Indeed, the poorest areas (those with the lowest values of AWI) are prone to experience higher NDP per disaster, while higher AWI is usually associated with lower NDP. Weather factors, especially precipitation levels, have a clear impact. More extreme hazards, characterized by high precipitation levels and strong wind speed, translate into more displacements (see Fig. 4, Fig. 5A, and Supple. information).

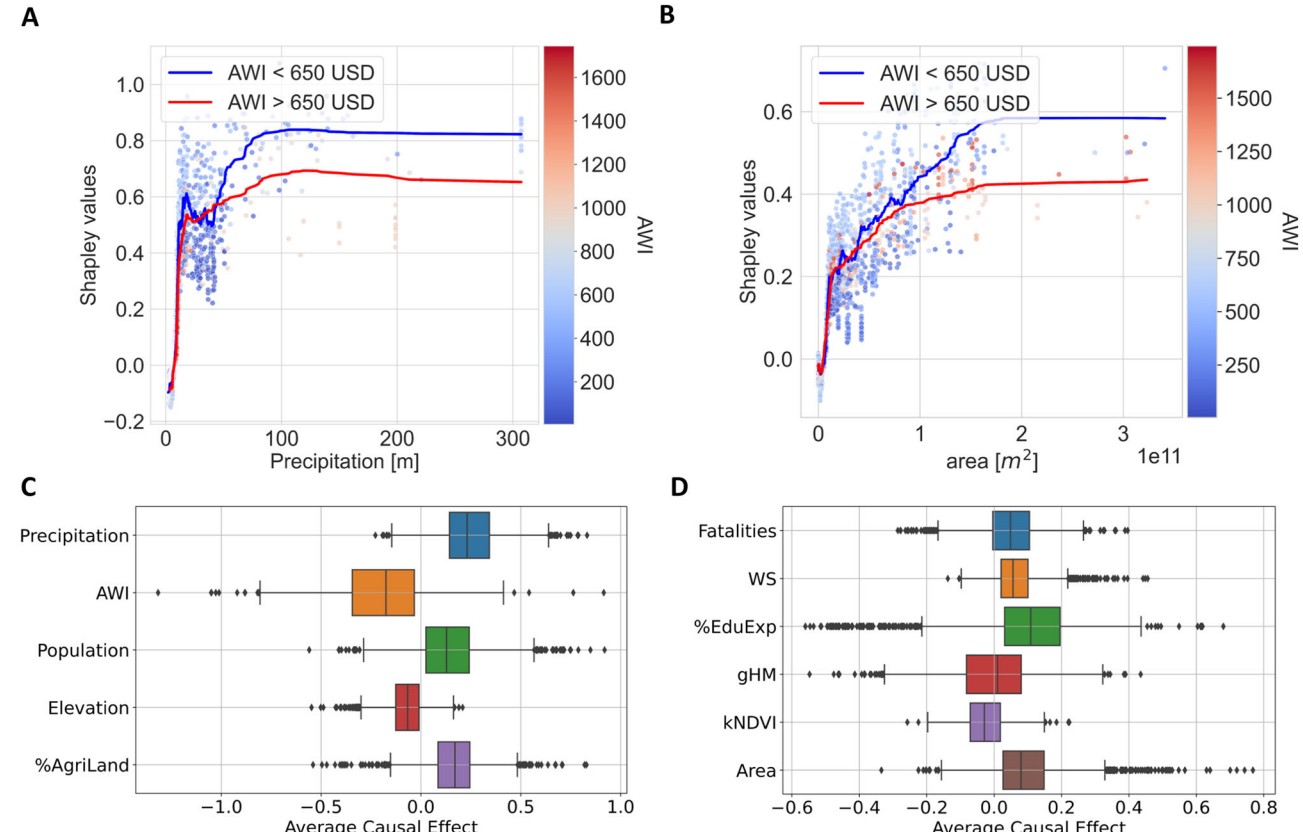

**Fig. 5 | Scatter plots of Shapley values versus precipitation (A) and area (B), and box plots of the treatment effects obtained with causal forests (C, D).** In the upper plots (**A**, **B**), the color scale is given by the value of the Absolute Wealth Index (AWI). The blue and red curves are smoothed averages of the Shapley values for instances having AWI < 650 US dollars and AWI > 650 US dollars, respectively. In the bottom graphs (**C**, **D**), we show the distribution (median and spread) of the causal relationship between the target (i.e., NDP) and each of the covariates considered one by one as treatments.

Interestingly, LR downplays the importance of precipitation and assigns greater importance to wind speed. In contrast, ML methods emphasize a stronger association between hazard and displacement; as highlighted by the Shapley values for both GBM and RF models, precipitation is consistently identified as one of the top two influential predictors. This can be attributed to the non-linear relationship between NDP and precipitation, further compounded by its interactions with vulnerability variables (see discussion below and also Fig. 5). Linear models are then insufficient for capturing these complex patterns, highlighting the necessity for ML approaches.

The third most important factor identified by the RF (second for the LR, and first for the GBM) is exposure given by the size of the affected area. All models predict a higher NDP when the affected area is more significant, as evidenced by Fig. 4 and 5B. Similar conclusions hold for the population covariate (see Supple. information), although it is classified as less important. This is likely due to the fact that it only considers human exposure, whereas affected area might also capture infrastructure and crop damage, which could worsen the severity of displacement. Numerous studies have revealed that violent conflicts serve as stressors. Similarly, critical situations such as weather-induced disasters can exacerbate these stressors, creating a vicious cycle[29,51,87–89]. Consistently, all models associate a higher NDP with a higher number of conflict fatalities (see Fig. 4, and Supple. information).

Regarding land-type exposure, Shapley values of the average elevation capture that high-altitude regions are less exposed since storms and flooding mainly hit coastal areas or villages around rivers (see Fig. 4 and Supple. Information), which are typically also the most densely populated. The models also capture that

agriculturally dependent countries usually are associated with higher NDP (see also Supple. information), which has been reported already in several works[90–93]. When livelihoods strongly depend on agriculture, weather hazards force people to move to seek other means of subsistence. According to all models, both gHM and kNDVI play a marginal role. However, to some extent, the fact that rural regions are more impacted can also be observed in their trends. Other human impacts on landscape captured by high gHM, such as infrastructures, electricity lines, and so forth, are a sign of more developed regions that are more resilient and less vulnerable (see also Supple. information). Events associated with higher kNDVI values, in turn, occur in more vulnerable areas, including cultivated fields, which are also among the most exposed[94,95]. Once exposed to a weather-related disaster, people's decision to leave is strongly influenced by adaptation, which is mainly driven by the skills of the affected community to diversify their income or even change their lifestyles. This is well-captured by the model as more NDP are generally associated with events in countries with lower education expenditures (cf. Fig. 4). Using data, we support previous expectations that higher investments in education could serve as an effective adaptation strategy[96–99]. Education works as a multiplier, as better-informed and risk-aware communities can be more resilient and more likely to adapt and react to environmental stressors. Furthermore, national education expenses may serve as proxies for other critical factors like governance quality and the effectiveness of a country's disaster response. These elements are crucial for a comprehensive understanding of vulnerability, resilience, and coping capacity, but obtaining them with adequate quality and coverage is often challenging[100].

To further confirm the claim of differential vulnerability, we focus on the interaction between AWI, precipitation, and area. In Fig. 5, we show the Shapley values of precipitation (A) and area (B) as a function of both weather and exposure, respectively, and AWI (more examples in Supple. information). Firstly, let us stress the non-linearity of the relation between NDP (quantified by the Shapley values) and both hazard and exposure factors. In particular, we observe an effect of saturation of the Shapley values in the high precipitation regimes and also for large areas but with a less steep growth, even if, at this stage, it is still not clear to what extent this reflects a property of the hazard-induced mobility phenomenon (e.g. due to the fact that the maximum impact is limited by the totality of the exposed elements) or possible biases in the input data (e.g., due to imperfect matching between polygons extracted and impacted area, see also Materials and methods). Then, we notice that the AWI does not have a discriminating power for events with lower precipitation and area. In contrast, it impacts events in the most extreme weather regime and with the largest affected areas. Hazards characterized by the same high levels of precipitation result in greater NDP when they occur in poorer areas with lower AWI (Fig. 5A). Insightful observations can also be obtained by looking at the intersection between vulnerability and exposure (Fig. 5B). Given areas with similar extents, more NDP occur when the AWI is lower, further confirming the hypothesis of a differential mechanism in place. Finally, to control for correlations and interactions among the covariates, we present in Fig. 5C and D the median of the causal effect of each predictor estimated using the causal forest algorithm[65]. The individual contributions are isolated by considering each covariate as a treatment. Its causal effect is determined by the variation in outcome conditioned on the remaining predictors[66] (see Materials and Methods and Supple. information for more details). Remarkably, the results match the Shapley values in showing which way the causal effect goes and which factors are most important. Still, it is worth noting that, based on the current dataset, none of the causal effects reach statistical significance (see also Supple. Information). The identification of these interactions has been made possible through the integration of ML models, which can capture intricate non-linear interdependencies, alongside XAI techniques and causal arguments. It's important to underscore that this approach uncovered data-driven patterns without necessitating any prior assumptions, and its effectiveness could be enhanced with the inclusion of more data over time.

## Discussion

Population movements induced by weather hazards are affecting millions of people globally, and this is expected to be exacerbated in the following decades, according to climate change projections[10]. Understanding the driving mechanisms is complicated because of the non-linear and largely unknown interactions between environmental, societal, and economic factors which traditional parametric models cannot capture[18,22,23,50]. To overcome the assumptions that limit mechanistic models, such as linear relationships or explicit functions for the interactions terms, we proposed data-driven machine learning techniques to model and explain human flows due to natural hazards from observational data alone. We focused on new internal displacements in the presence of sudden-onset disasters and exploited XAI and causal methods to unravel the main drivers of the phenomenon. A displacement dataset at a sub-national level was presented. The models identified structural factors that dominated the magnitude of movements and highlighted the relevance of socioeconomic conditions and hazard exposure factors. Relying solely on data, we showed that variables related to weather hazards were useful predictors and that the amount of NDP depends on the interactive effects of precipitation and local wealth status. These findings match

previous studies[53,78–81,101–103] reporting that the impact of environmental stressors on displacement is crucially interconnected with the socio-economic conditions of the affected area as well as with its exposure and additional exacerbating factors like the presence of conflicts.

Alternative ways of characterizing the hydro-climatic dimension of the phenomenon should be considered. Indeed, choosing weather variables with high predictive power is non-trivial since, in many cases, people's movements occur even in non-anomalous weather conditions, and lagged effects over longer time scales can also be present. In this regard, obtaining geolocated displacement data is crucial to advance research in human mobility studies. Even when approximate information on the location is present, there is no unique way of defining the affected area. Thus, additional efforts should be made to improve the identification of the polygons of interest with the highest possible accuracy. Other data gaps must also be addressed as the information available is often limited to aggregated levels, typically on a national scale. Further components, such as the coping capacity, could also be considered to improve the characterization of human displacement risk. Ultimately, to make substantial progress, high-quality and high-resolution variables are required. However it is also important to acknowledge that, while this study employs a top-down analytical approach to examine displacement flows, people movement decisions are the result of multiple individual considerations, which may not be entirely captured by quantitative variables within the scope of statistical analysis.

In conclusion, the concept of differential vulnerability was evidenced, inferred, and quantified by the machine learning model from observational data alone, without assumptions or preconceived relations. In this way, hypotheses, expectations, and qualitative analyses by domain experts found further empirical confirmation. Most importantly, XAI allowed us to shed light on the intricate interplay among the three dimensions of disaster risk, overlooked in conventional multi-hazard risk models, which often treat hazard, exposure, and vulnerability as independent components. Given that, our study and methodology can be a stepping stone for advancing evidence-based mitigation strategies and policies in the future, capitalizing on current strides in both modeling techniques and data accessibility.

## Methods

### Building a global dataset of displacements

For each storm, flood, or landslide in the years 2016–2021, the target is given by the number of NDP as reported in the Global Internal Displacement Database by IDMC[61] for that specific event. "New Displacement" refers to the number of new cases or incidents of displacement recorded over the specified event[104] From the names of the impacted location, geo-referenced polygons were extracted with the OSMnx Python library, which is based on OpenStreetMap[71]. In some cases, a polygon was not found. Thus, the smaller matching administrative level area, including the affected region, was considered. This can introduce bias in the variables extracted at polygon resolution since the polygon would be greater than the affected area. However, polygons with areas covering entire countries were discarded to reduce such aggregation bias. Areas for each polygon were calculated by projecting the polygon into the UTM CRS zone where its centroid lies. NDP data were integrated and harmonized with satellite-derived variables, weather information, and socioeconomic data to construct the modeling input database. In particular, Google Earth Engine (GEE) was used to extract the weather covariates within the polygons. The values of 10m wind speed in $m/s$ ($v$ and $u$ component) and precipitation in $m$ are obtained from the ERA5-Land hourly dataset from the Latest Climate Reanalysis Produced by ECMWF by the Copernicus Climate Change Service[67] from the GEE repositories. They have hourly temporal resolution and a spatial resolution of about 9 km. The total wind speed was computed as the square root of the sum of the squares of $u$ and $v$. Wind Speed, precipitation, and kNDVI variables

had to be temporally and spatially aggregated, and the reported period of each disaster is 20 days on average. For wind speed, the maximum value over the hazard duration and each polygon were considered. For precipitation, the sum over the hazard duration and the maximum value of this sum per polygon was computed (i.e., maximum precipitation accumulation). An analogous procedure was followed to aggregate the spatial and temporal mean of kNDVI[105], elevation[70], population counts[72] and gHM[74], characterizing vegetation dynamics, topography, human exposure, and proxy to the anthropogenic action on land respectively. The AWI was derived from the Relative Wealth Index from Meta's Data4Good[73], a finely-grained sub-national poverty index combining connectivity, satellite, and household survey information. The index is available in a 2.4 km grid. The aggregation was conducted by intersecting the grid with the disaster polygons and taking the maximum index values. The area[71] of the polygons in $m^2$ was added as a covariate to provide an approximate characterization of exposure not only in terms of persons but also of exposed assets, buildings, facilities, infrastructures, and so on. The Education Expenditure (as a percentage of the GNI) and Percentage of Agricultural Land over a country were collected from the United Nations Statistics Division SDG API[69]. The last year's value was considered when the indicator's value was missing for a specific year. We introduced the conflict dimension of displacement by taking the annual sum of fatalities resulting from conflict events over a polygon from the ACLED[75] database. This completed the harmonization of the database at a disaster level. Note that we included only the countries for which a harmonized database for the selected variables could be constructed; therefore, not all countries from the IDMC database could be included in the study. Currently, AWI data is exclusively accessible for low-middle-income countries[73]. This leads to a dataset that is uniform in its composition for analysis. However, it may introduce a bias based on income when attributing the impact of hazards on a global scale. Furthermore, we acknowledge that certain chosen variables may act as proxies, potentially correlating with other underlying factors like the effectiveness of disaster management or the community's coping capacity. Additionally, there is a potential for confounding effects among these variables, which we have partially investigated through a predominantly data-driven approach. This highlights the importance of thoroughly addressing any data gaps associated with potentially explanatory variables and processes. These aspects, though often challenging to observe, hold significance in comprehending the phenomenon. Such efforts are essential to guarantee that the sufficiency assumption is adequately met. The predictors and data granularity are summarized in Table 1.

## Data pre-processing and machine learning model training

Data was standardized via a $z$-score procedure (subtract the mean and divide by the standard deviation). To account for the skewness of the distribution and reduce the weight of outliers, the target was computed as the logarithm of the NDP. We also tried an alternative choice for the target by re-scaling the number of NDP by the population of the polygon without taking the logarithm. Still, the performance of the RFs was much lower. The same log transformation was applied to the total population variable and conflict fatalities before scaling all covariates. We implemented both GBM[63] and RF[62] regression models to estimate NDP from predictors in Table 1. RFs are nonparametric models that do not assume any particular structure on the data and can capture nonlinear relationships and interactions between the covariates. They are also suitable for heterogeneous input variables. RF models can handle high-dimensional problems while minimizing the risk of overfitting. They do this by combining many trees operating with different feature subsets randomly picked. Through the so-called recursive partitioning, RF builds a decision tree out of the strongest available predictors. The RF method does this repeatedly and then averages

all the decision trees together to make a prediction[62]. GBMs are ensemble methods that sequentially combine weak learners by minimizing the error made by the previous ensemble at each step[63]. For a standard machine learning task, the Global Internal Displacement Database is of limited size. For this reason, bootstrapping with ~$10^3$ iterations was used to estimate the statistical scores, namely average accuracy, RMSE, ME, and goodness-of-fit $R^2$[106]. Both RF and GBM models were fitted using stratified sampling with 70–30% train-test partitions. To stratify the splitting, we employed the quantile binning strategy on the target variable to include similar fractions of NDP values in both test and train data. All RFs had the same hyper-parameters: 'maximum depth of the trees' was set to 6, 'the minimum number of samples in a node to split' to 4, the 'maximum number of features' to 3, and 'the number of trees' to 40. All GBMs had the same hyper-parameters: the 'number of estimators' was set to 60, the 'minimum samples per split' to 4, the 'minimum samples per leaf' to 2, the 'maximum depth' to 4, and the 'learning rate' to 0.05. These hyper-parameters were tuned by following a grid search to reduce overfitting defined as one minus the ratio between $R^2_{test}$ and $R^2_{train}$. The same training and testing procedures and specifications were used with and without the weather variables. The code for the machine learning modeling is mainly based on the Python library scikit-learn and is available at https://github.com/IPL-UV/AI4Migrations. See also Supple. Information for additional details and other experiments performed using different cross-validation strategies as well as subsets of the data based, e.g., on the continent or hazard type.

## Shapley values, causal treatment effects, and individual conditional expectations

Shapley values were used to estimate how the input features affect the predicted target[83]. They were first introduced in the context of game theory[84]. Now, they are one of the most used XAI techniques for ranking the input features and estimating their contribution to the model's predictions per instance. The total gain (i.e., the prediction) is divided between the players (i.e., the covariates) by considering all possible coalitions that can form and calculating the (average) change in the outcome. In this way, the importance of each predictor is properly weighted by considering the interactions between input features. All computations and experiments were done with the python package SHAP[85]. See Supple. information for additional details on the formalism and theory. The causal forest model[65,66] is specifically designed to estimate conditional average treatment effects. To claim a causal link, a treatment must produce a change in outcome while all other covariates are held constant; this type of treatment-induced change is known as an intervention. The average treatment effect refers to the effect of a treatment on the outcome, taking into account the other covariates. For each observation, the model can predict two potential outcomes using two conditional mean functions, one for the treatment group and one for the control group. The difference between these potential outcomes represents the average causal effect. All experiments have been performed using the EconML package[107]. More information on causal forests and double ML are in Supple. information. Finally, individual conditional expectation (ICE) plots illustrate how a prediction varies for each instance when a specific feature is changed. To generate ICE plots[108,109], we keep all other features constant while producing variants of the instance by substituting the selected feature's value with a set of values from a grid. These newly generated instances are then used to create predictions with the model, resulting in a set of points for each instance with the feature value from the grid and the corresponding predictions. ICE plots can provide further complementary insights, particularly in scenarios where interactions between the features are present. Additional technical details and results are presented in Supple. information.

**Reporting summary**

Further information on research design is available in the Nature Portfolio Reporting Summary linked to this article.

## Data availability

All data needed to support the conclusions in the paper are freely available at https://www.internal-displacement.org/sites/default/files/UVEG_IDMC_global_dataset_natcomm.xls. The harmonized dataset will be regularly updated and maintained in collaboration with IDMC. Displacement figures can be found at https://www.internal-displacement.org/database/displacement-data. AWI data is available at https://dataforgood.facebook.com/, and all the other covariates can be downloaded from the links in the references. The analysis-ready dataset used for this study can be downloaded at: https://zenodo.org/records/10063853.

## Code availability

The code and some demos containing our experiments can be found in https://zenodo.org/records/10063853.

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

## Acknowledgements

The authors thank Markus Reichstein, Miguel Mahecha, Nuno Carvalhais and Ghjulia Sialelli for commenting on an earlier version of the manuscript. M.R. and J.M. thank Alex Pompe, Guanghua Chi, and Eugenia Giraudy for discussions about AWI data. G.C.V. would like to acknowledge the support from the European Research Council (ERC) under the ERC Synergy Grant USMILE (grant agreement 855187), the support of the Fundación BBVA with the project 'Causal inference in the human-biosphere coupled system (SCALE),' and the European Union's Horizon 2020 research and innovation program within the project 'XAIDA: Extreme Events - Artificial Intelligence for Detection and Attribution,' under grant agreement No 101003469. J.M., Q.W., M.P., G.C.V., and M.R. thank the support of the European Union's Horizon 2020 research and innovation program within the project 'DeepCube: Explainable AI pipelines for big Copernicus data' (grant agreement No 101004188). E.S.M. thanks the support of the H2020 ELISE - European Network of AI Excellence Centres - research and innovation programme under grant agreement No 951847.

## Author contributions

M.P., G.C.V., and J.M.T. conceptualized the study, whose early version was presented at the NeurIPS workshop on Machine Learning for Humanitarian Disaster Management in 2020[103]. M.R. designed the analysis, contributed to data preparation and preprocessing, and carried out all the statistical modeling and the XAI experiments. J.M.T. performed the data processing and harmonization. J.M. contributed to the design and ran some of the experiments. M.T.E., S.P and E.S.M. contributed to the definition and setup of the framework and provided displacement data. J.M., E.S.M., S.P., M.T.E., and Q.W. provided crucial comments on the first versions of the manuscript. M.R., J.M.T., and G.C.V. prepared the manuscript with contributions from all co-authors. All authors reviewed the manuscript.

## Competing interests

The authors declare no competing interests.
