## [Peer Review File · Nature Communications]

Exploring interactions between socioeconomic context and natural hazards on human population displacementReviewers' Comments:

Reviewer #1:

Remarks to the Author:

This paper studies how different covariates are related to the number of newly displaced persons (NDPs) after different types sudden-onset hazards. Specifically, the main questions the paper is looking to answer are "what conditions and combinations of drivers give rise to higher NDPs?" and does the differential vulnerability hypothesis hold in this setting? To answer this the authors:

- prepare a dataset of the number of NDPs over 2399 disaster events in the period 2016–2021 with economic, weather, and land specific covariates, including: precipitation, wind speed, nonlinear normalized difference vegetation index (kNDVI), global human modification (gHM) index as a measure of the anthropogenic action on land, elevation, education expenditures (%EduExp), Absolute Wealth Index (AWI), fraction of agricultural land (%AgriLand), and population.
- Model $\log(\text{NDPs})$ with the above covariates using a random forest model and analyze the resulting SHAP values to draw conclusions about the main questions

The results include many interesting relationships, such as that poorer areas experience higher numbers of NDPs per disaster, and that more intense hazards result in larger numbers of NDPs (and differentially so for poorer areas!)

Comments

Firstly, this is a very well written and put together paper with a thorough analysis and discussion of results. My main concerns with it are twofold

- Modeling approach: the paper is trying to answer a causal question, "what conditions and combinations of drivers give rise to higher NDPs?", with observational data. The more confounders that the modeling process can control for, the more convincing the results will be. With this in mind, the authors should consider experimental setups that use geographic splits (vs 70/30 random splits) and (perhaps separately) temporal splits. If the same relationships hold under these settings, for instance, then this is better evidence that it is causal. Separately, the authors might consider other models (e.g. XGBoost or EBMs) that might fit the regression task better.
- Baseline comparisons: adding a simple linear regression model as a baseline would greatly strengthen the paper. The paper argues that the relationships are non-linear, but how well are they captured by a linear model? I.e. in Table 1 if the R^2 of a linear model is much lower than the RF, then it makes sense to study NDPs with non-linear models and XAI methods. If not, then do the same conclusions still hold with the linear model?

I would recommend that this paper be revised with the above experiments and resubmitted.

Minor questions/comments

- The "growing frequency and severity of weather extremes" is stated as a given, is there a citation for this?
- The hyperparameters for the RF model seem to be tuned (or far from the defaults of the scikit-learn package in any case). Is there a reason for this? Can this be explained in the paper?
- Will the dataset be released publically? (zenodo.org is a great choice for this)
- The area of the disaster polygon would be an interesting covariate (and would definitely help as a control)

Reviewer #2:

Remarks to the Author:

This article uses machine learning to study the role of environmental, societal, and economic factors in driving internal displacement during floods, storms, and landslides from 2016-2021. The authors compile a dataset of diverse covariates at the global level and then attempt to understand the relative role of hazard, exposure, and vulnerability in driving forced displacement. Overall this is an ambitious piece of research that tackles an interesting problem. The dataset which the authors have compiled is alone valuable, and I appreciated the effort to use machine learning explainability techniques to understand interactions between different displacement drivers.

Nevertheless, I was not fully convinced by the authors' technical approach and had a few key questions:

(1) My biggest concern is that the authors do not clearly distinguish between correlation and causation. The authors use SHAP values and variable importance weights to determine the role of different factors in driving displacement, making statements such as "higher AWI leads to lower NDPs" and "more intense hazards translate into more displacements". However, the RFs are designed to make accurate predictions, rather than infer causal relationships – the latter would at minimum require further reasoning by the researcher interpreting the findings, or preferably the use of causal methods (see e.g. Wager and Athey's Causal Forests). Mullainathan and Spiess (Journal of Economic Perspectives 2017) provide a very nice discussion of the issues with using standard machine learning algorithms for causal inference, but effectively, the problem is that if two variables are highly correlated with each other (for example, AWI and education expenditures), then they can act as substitutes in machine learning approaches like random forests that use some sort of variable selection or regularization, which can complicate the interpretation of variable importance weights and mean that the selection of one at the expense of another is not particularly meaningful. I think the authors need to better understand and control for the correlations between their explanatory variables, or at least use more careful language and explicitly list this as a limitation.

(2) Second, the authors make some technical choices that do not seem particularly well motivated. For example, they normalized their feature variables and log-transformed the population variable, but I don't think that these transformations would affect the performance of a partition-based method like random forests. Similarly, they normalize the target variable by taking the log transformation, but they don't address the consequences of this decision – which effectively means that they are de-prioritizing the accurate prediction of the size of large-scale displacements (relative to using raw target values). They note the "saturation of the Shapely values" and speculate about causes of this, but I wonder if this saturation might actually emerge as a consequence of the transformation of the target values? Finally, the authors say that they conduct bootstrapping because of the small sample size, but the sample size does not appear particularly small to me, and given the randomness already inherent in the RF algorithm I am not entirely sure what is being gained by this. Perhaps it might be worth looking into existing methods for uncertainty quantification / confidence interval creation for RFs instead? (e.g. <https://contrib.scikit-learn.org/forest-confidence-interval/>)

(3) Third, the authors do not discuss some of the other choices they made. For example, how were the hyperparameters of the RF algorithm chosen? Was cross-validation used? Why was the RF algorithm chosen in the first place, relative to other methods? The number of variables in the model is actually fairly small, so I feel that even linear regression or basic correlation analysis could have yielded some of the key insights here. As the authors note, random forests would be ideal for handling large numbers of variables, but it looks like a lot of the variable selection was done by the researchers prior to fitting the models presented here ("a wide variety of features were considered and, among them, those with the most predictive power and also better semantic meaning were selected").

In general, these seem like shortcomings of the paper that could be addressed by a major revision, but on investigating a question I had about a statement in the supplementary material it appears that parts of the section on Shapely values have been taken verbatim from another source without citation. This casts doubt on the ethical integrity of the remainder of the article and its findings and based on

this I would not recommend the article for further consideration.

Reviewer #3:

Remarks to the Author:

Exploring interactions between societal context and natural hazards on human population displacements

This paper uses a data-driven approach with machine learning methods to predict internal displacement as a result of natural hazards around the world. The paper then investigates the primary predictors of such displacement, including variables related to vulnerability. The dataset compiled is very impressive, and the figures are excellent! I believe that the work is both relevant and timely. A key result is that weather variables (precipitation especially) improve the model's predictive ability of displaced population. This suggests that, to some extent, environmental conditions are important for predicting displacement globally.

Major comments

- I recommend restructuring/ strengthening the introduction. For example, critical concepts such as vulnerability, displacement, and hazard exposure should be clearly defined and introduced up front. Especially vulnerability, which is a key component of this work, should be explained early in the paper.
- More justification of how and why the variables were selected should be included in Methods. For example, why not include data on temperature extremes? Why were hazards limited to floods, storms, and landslides? Droughts, for example, have been shown to be very important for migration, especially in rural and agricultural areas.
- It is mentioned in future work, but I think that it might be important at this stage of research to include conflict as a variable in the model.

Minor comments

Introduction

- In the second sentence, suggest removing "one of the main" and instead say that migration is one possible adaptation strategy
- Can you clarify what you mean by "makes it difficult to implement effective measures"? Effective measures for what?
- You should introduce what you mean by vulnerable/ vulnerability before you claim "All of this aggravates the burden on the most vulnerable areas." Also what burden? I think more clarity/ specificity would help.
- Remove "certainly" in second paragraph
- Why are environmental perceptions important? (Introduced in second paragraph)
- What kind of insecurity are you referring to at the end of the second paragraph?
- More generally- You might introduce the concept of forced displacement sooner/ restructure the introduction so that it is clearer. You mention it in the first and second paragraphs before really introducing what it means as a concept
- Can you clarify what "three dimensions of the problem" you mean in the third paragraph?
- Hazards, exposure, and vulnerability are three very specific, distinct concepts that deserve being introduced / defined up front
- Can you cite the sentence about how adaptive capacity and economic resources in a community matter for displacement?
- Is it true that the majority of research focuses on international migration? How do you know? If it's not known, then perhaps just say that much research focuses on international migration
- Can you cite the sentence about data limitations?

Results

- Not sure what you mean by "factorial complex problem"

- Can you very briefly state what the sufficiency assumption is?
- Why did you pick these variables?
- A brief explanation/ justification of why you selected RF rather than another algorithm is warranted.
- Can you more clearly motivate why you limit the analysis to low and middle income countries (also as defined by who)? I think introducing the concept of vulnerability more clearly in the intro, as mentioned, might help this
- Is your time scale a year for the data?
- Why did you select the thresholds that you did for Figure 5? (600 USD and 10m precipitation?)

Discussion

- You mention this as a next step, but I think it would be important to include conflict as a predictor for this work. It would be very interesting to know how strong of a predictor conflict is/ how your model performance improves.

General

- I suggest citing this source as an example of using RF to study environmental migration- Best KB, Gilligan JM, Baroud H, Carrico AR, Donato KM, Ackerly BA, Mallick B. Random forest analysis of two household surveys can identify important predictors of migration in Bangladesh. Journal of Computational Social Science, 2020.

Notes on revision made to manuscript Paper # NCOMMS-22-40518

Response to the reviewers

Reviewer 1

Reviewer Comment 1.1 — This paper studies how different covariates are related to the number of newly displaced persons (NDPs) after different types sudden-onset hazards. Specifically, the main questions the paper is looking to answer are "what conditions and combinations of drivers give rise to higher NDPs?" and does the differential vulnerability hypothesis hold in this setting? To answer this the authors: - prepare a dataset of the number of NDPs over 2399 disaster events in the period 2016 – 2021 with economic, weather, and land specific covariates, including: precipitation, wind speed, nonlinear normalized difference vegetation index (kNDVI), global human modification (gHM) index as a measure of the anthropogenic action on land, elevation, education expenditures (%EduExp), Absolute Wealth Index (AWI), fraction of agricultural land (%AgriLand), and population. - Model $\log(\text{NDPs})$ with the above covariates using a random forest model and analyze the resulting SHAP values to draw conclusions about the main questions

The results include many interesting relationships, such as that poorer areas experience higher numbers of NDPs per disaster, and that more intense hazards result in larger numbers of NDPs (and differentially so for poorer areas!)

Reply: We are deeply grateful to the reviewer for the very positive feedback and valuable comments that have improved the quality of this paper. An item-by-item response is provided below.

Reviewer Comment 1.2 — Firstly, this is a very well written and put together paper with a thorough analysis and discussion of results. My main concerns with it are twofold - Modeling approach: the paper is trying to answer a causal question, "what conditions and combinations of drivers give rise to higher NDPs?", with observational data. The more confounders that the modeling process can control for, the more convincing the results will be. With this in mind, the authors should consider experimental setups that use geographic splits (vs 70/30 random splits) and (perhaps separately) temporal splits. If the same relationships hold under these settings, for instance, then this is better evidence that it is causal. Separately, the authors might consider other models (e.g. XGBoost or EBMs) that might fit the regression task better.

Reply: To strengthen the results of our work and reduce the possibility of confounding effects, we performed experiments with geographical splits (i.e. labelling events by country) and temporal splits (i.e. labelling events by year and month). The outcomes of these additional explorations have been included in the Supp. info. in the section *Cross-validation: spatial and temporal splits* on page 22. We notice that the temporal sampling does not change significantly the performance of the RF, while there is a degradation in the case of country-based splits. As we discuss in the paper, this can be explained by the fact that data are not evenly distributed among countries (see Fig. 14). Nonetheless, in both setups the Shapley values maintain the same qualitative trends and relationships, thereby supporting

the conclusions discussed in the main text. Moreover, we also fitted GBM models and their performance is comparable to that of the RFs, see Table 1. We also compare the Shapley values of the GBM with those of the RF. Despite some minor differences in the ordering of the features, both ensemble methods learn similar relationships between NDP and the chosen drivers, see Fig. 4. Finally, to provide some preliminary evidence of the causal links between NDP and the predictors, we also computed the average causal effect of each feature on the NDP and results are summarized in Fig. 5. See also the section *Causal Forest: theory, properties and code* on page 19. Along the same line, we included the individual conditional expectation plots (See Fig. 11, 12, 13) in the section *Individual conditional expectations: theory, properties and code* on page 17. Overall, the ICE plots also confirm the findings obtained with Shapley values.

Reviewer Comment 1.3 — Baseline comparisons: adding a simple linear regression model as a baseline would greatly strengthen the paper. The paper argues that the relationships are non-linear, but how well are they captured by a linear model? I.e. in Table 1 if the R^2 of a linear model is much lower than the RF, then it makes sense to study NDPs with non-linear models and XAI methods. If not, then do the same conclusions still hold with the linear model?

Reply: We had already tried a simple linear regression but the results had not been included in the previous version of the manuscript. We have now added them in the main text, see Table 1 and Fig. 4. We thank the reviewer for pointing this out since this definitely helps us justifying the need for more complex ML models. Indeed, the R^2 of the linear regression model is much lower than that of both RF and GBM. Moreover, it is also interesting to notice that the linear model fails to capture the importance of the relationship between precipitation and NDP (see Fig. 4A), which is instead well captured by ensemble ML models (Fig. 4B, 4C).

Reviewer Comment 1.4 — I would recommend that this paper be revised with the above experiments and resubmitted.

Minor questions/comments:

- The "growing frequency and severity of weather extremes" is stated as a given, is there a citation for this? The hyperparameters for the RF model seem to be tuned (or far from the defaults of the scikit-learn package in any case). Is there a reason for this? Can this be explained in the paper?
- Will the dataset be released publically? (zenodo.org is a great choice for this)
- The area of the disaster polygon would be an interesting covariate (and would definitely help as a control)

Reply: We added all the experiments asked by the reviewer. Here the answers to his minor comments:

- To support the statement on the rise in the frequency and severity of the hazards we added Refs. 6-10, among which the IPCC report.
- The hyperparameters of both RF and GBM were tuned using a finite grid search. We added a sentence in the Methods section on page 8.
- The dataset will be made fully public when the paper will be published. However, we already added in the text a link to the full dataset which the reviewers can download (see Data Availability section on page 8). We kindly ask though to keep it private and avoid sharing until the publication of our work.

- We added the area among the covariates (see Table 2) and this has improved the models’ performance. Moreover, we find that the area gives a better characterization of the exposure with respect to the population variable. See the discussions in the main text, Methods and Supp. information.

Reviewer 2

Reviewer Comment 2.1 — This article uses machine learning to study the role of environmental, societal, and economic factors in driving internal displacement during floods, storms, and landslides from 2016-2021. The authors compile a dataset of diverse covariates at the global level and then attempt to understand the relative role of hazard, exposure, and vulnerability in driving forced displacement. Overall this is an ambitious piece of research that tackles an interesting problem. The dataset which the authors have compiled is alone valuable, and I appreciated the effort to use machine learning explainability techniques to understand interactions between different displacement drivers.

Reply: The reviewer’s insightful comments have been immensely helpful in improving this paper. We are deeply grateful for the positive feedback that has guided us in refining our work to a great extent. By incorporating the reviewer’s suggestions, we have not only improved the overall quality of the paper but also made our results much more solid. We provide an item-by-item response below.

Reviewer Comment 2.2 — (1) My biggest concern is that the authors do not clearly distinguish between correlation and causation. The authors use SHAP values and variable importance weights to determine the role of different factors in driving displacement, making statements such as “higher AWI leads to lower NDPs” and “more intense hazards translate into more displacements”. However, the RFs are designed to make accurate predictions, rather than infer causal relationships – the latter would at minimum require further reasoning by the researcher interpreting the findings, or preferably the use of causal methods (see e.g. Wager and Athey’s Causal Forests). Mullainathan and Spiess (Journal of Economic Perspectives 2017) provide a very nice discussion of the issues with using standard machine learning algorithms for causal inference, but effectively, the problem is that if two variables are highly correlated with each other (for example, AWI and education expenditures), then they can act as substitutes in machine learning approaches like random forests that use some sort of variable selection or regularization, which can complicate the interpretation of variable importance weights and mean that the selection of one at the expense of another is not particularly meaningful. I think the authors need to better understand and control for the correlations between their explanatory variables, or at least use more careful language and explicitly list this as a limitation.

Reply: We thank the reviewer for stressing the role of causality and its differences with respect to correlations. These comments motivated us to perform additional experiments which have strengthened the results of our paper. First of all, we recognise that RFs only optimise the loss function and, thus, they can learn in some cases also spurious relationships. This is the reason why we then apply multiple XAI methods which allow us to extract the learned patterns and check whether they make sense and are interesting from a domain-knowledge perspective. However, following reviewer’s comments, in the

revised version we have added the results obtained with the causal forest model. The estimation of the average causal effect of each input feature is presented in Fig. 5. Remarkably, the average treatment effects of the most important variables (i.e. precipitation, AWI, and area) confirm what we had already found with the Shapley values. We added a section in Supp. info. where we discuss in more detail the formalism of double ML and causal forest (see pages 19 and 20). The fact that the causal analysis does not modify the results we had obtained with RF combined with XAI can be explained by the low correlations among the used covariates (see e.g. Fig. 19), which have been chosen to be as much orthogonal as possible. To give further support to our conclusions, we also included the ICE plots in the Supp. info. (see pages 19, 20, 21). Here, for most of the features, the curves for the different instances follow similar paths suggesting that there are limited or no interactions among the covariates. At the same time, we are aware of the difficulty (if not impossibility) of controlling for all confounding factors in observational studies. For this reason, we do not claim to have discovered causal links.

Reviewer Comment 2.3 — (2) Second, the authors make some technical choices that do not seem particularly well motivated. For example, they normalized their feature variables and log-transformed the population variable, but I don't think that these transformations would affect the performance of a partition-based method like random forests. Similarly, they normalize the target variable by taking the log transformation, but they don't address the consequences of this decision – which effectively means that they are de-prioritizing the accurate prediction of the size of large-scale displacements (relative to using raw target values). They note the “saturation of the Shapely values” and speculate about causes of this, but I wonder if this saturation might actually emerge as a consequence of the transformation of the target values? Finally, the authors say that they conduct bootstrapping because of the small sample size, but the sample size does not appear particularly small to me, and given the randomness already inherent in the RF algorithm I am not entirely sure what is being gained by this. Perhaps it might be worth looking into existing methods for uncertainty quantification / confidence interval creation for RFs instead? (e.g. <https://contrib.scikit-learn.org/forest-confidence-interval>)

Reply: We thank the reviewer for his comments and justify here our technical choices. The normalization does not indeed affect the performance of the RF, but it is applied mainly to have all covariates on a similar scale in such a way that it is then possible to compare the Shapley values (see e.g. Fig. 4). The variables are heterogeneous and have completely different units so we found useful to rescale them in order to be able to make direct quantitative comparisons. Log normalization is useful to deal with skewed or kurtotic distributions of continuous variables. We are aware that in this way we give less importance to very large values of displacement, and this is done to reduce the role of outliers. Moreover, this has a clear (and positive) impact on the results of the RFs (as well as makes training far easier and more stable). In fact, without taking the log transformation the R^2 goes down to less than 0.1, i.e. much lower than what we get by modelling the logarithm of NDP. Finally, log transformation represents quite a common choice when modelling counts, such as the number of displaced people, and it is used a lot in migration studies (see e.g. Ref. 54). We also checked that some degree of saturation of the Shapley values is present also without making the logarithm transformation. Let us mention that such an effect can be expected for some variables. For instance a trivial plateau is given by the total population in a given area since, at most, the number of displaced persons will equal the people living in the area. Finally, bootstrapping is needed to have an estimate of the performance of the models taking into account that we have small data that are context (and event) dependent. Even if there is not a precisely defined size (it actually depends on the complexity of the problem under study), typically

ML datasets have tens of thousands of records, while we only have less than 3000 events. We know that there are other ways of estimating confidence intervals but bootstrapping is also a very common technique (see e.g. <https://pubmed.ncbi.nlm.nih.gov/10797513/>).

Reviewer Comment 2.4 — (3) Third, the authors do not discuss some of the other choices they made. For example, how were the hyperparameters of the RF algorithm chosen? Was cross-validation used? Why was the RF algorithm chosen in the first place, relative to other methods? The number of variables in the model is actually fairly small, so I feel that even linear regression or basic correlation analysis could have yielded some of the key insights here. As the authors note, random forests would be ideal for handling large numbers of variables, but it looks like a lot of the variable selection was done by the researchers prior to fitting the models presented here (“a wide variety of features were considered and, among them, those with the most predictive power and also better semantic meaning were selected”).

Reply: We thank the reviewer for pointing this out and modified the text accordingly to clarify all the choices we made. The hyperparameters have been selected by using a simple grid search, as we now explain in the Methods section on page 8. In the Methods section we also explain how we estimated the metrics (i.e. ME, RMSE and R^2) by using bootstrapping, see also Fig. 3. Additional experiments using temporal (i.e. based on year and month values) and spatial (i.e. based on country) cross validation have been added in Supp. info. as suggested by Reviewer 1. We had previously tried other models, such as linear regression, Poisson regression, Gaussian Processes, shallow neural networks, but RFs performed better and are particularly suitable for heterogeneous tabular data. However, we have now added also the results obtained with a baseline linear regression and with GBM models, see Table 1 and Fig. 4. A table with the Pearson correlations among the predictors and also the target has been added in Fig. 19.

Reviewer Comment 2.5 — In general, these seem like shortcomings of the paper that could be addressed by a major revision, but on investigating a question I had about a statement in the supplementary material it appears that parts of the section on Shapely values have been taken verbatim from another source without citation. This casts doubt on the ethical integrity of the remainder of the article and its findings and based on this I would not recommend the article for further consideration.

Reply: We apologize for this inconvenience. The review part on the Shapely values in Supp. info. was not original and had been written by one of us, while the rest of the authors did not realize that the source had not been cited. We added Ref. 86 as recommended by the Editor. We have also heavily rephrased the material included there (see all changes in blue color). In addition, and to remove any shadow of a doubt, we share the data:

https://www.internal-displacement.org/sites/default/files/UEVEG_IDMC_global_dataset_natcomm.xls; and code on the dedicated web page of the paper on GitHub: <https://github.com/IPL-UV/AI4Migrations>.

Reviewer 3

Reviewer Comment 3.1 — This paper uses a data-driven approach with machine learning methods to predict internal displacement as a result of natural hazards around the world. The

paper then investigates the primary predictors of such displacement, including variables related to vulnerability. The dataset compiled is very impressive, and the figures are excellent! I believe that the work is both relevant and timely. A key result is that weather variables (precipitation especially) improve the model's predictive ability of displaced population. This suggests that, to some extent, environmental conditions are important for predicting displacement globally.

Reply: We are very happy for the excitement of the reviewer with respect to our work. In particular, we are pleased to hear that the reviewer appreciated the figures we produced to summarize the main findings and recognized the significance of the novel dataset we harmonized and presented in the paper. We are grateful for the reviewer's questions and comments, as they have helped us to clarify and improve several aspects of the analysis and the text. We have provided detailed answers below to address each comment.

Reviewer Comment 3.2 — Major comments

- I recommend restructuring/ strengthening the introduction. For example, critical concepts such as vulnerability, displacement, and hazard exposure should be clearly defined and introduced up front. Especially vulnerability, which is a key component of this work, should be explained early in the paper.

Reply: The introduction has been largely restructured by following the suggestions of the reviewer. In particular, the concepts of displacement, vulnerability and exposure are now defined in the first two paragraphs on page 1. Then, a more specific definition of NDP is given in the last paragraph of page 2.

Reviewer Comment 3.3 —

- More justification of how and why the variables were selected should be included in Methods. For example, why not include data on temperature extremes? Why were hazards limited to floods, storms, and landslides? Droughts, for example, have been shown to be very important for migration, especially in rural and agricultural areas.

Reply: Thank you for your comment. We chose to focus on sudden-onset weather extremes for many reasons. One main reason is their shorter time scale, typically ranging from days to a few weeks, while droughts can last for several years. Including droughts would require a different data aggregation method in terms of both time and space, which was not feasible for this study. By restricting our analysis to short time scales, we also reduced the impact of possible confounding factors that may be present over longer periods, such as policies, human interventions, and other stressors. Moreover, slow and sudden-onset disasters have different dynamics and mechanisms, making it challenging to model them with a single ML approach. We also investigated the peculiarities of each hazard type separately in the Supplementary Information, as shown in Figure 7. We provide further details on the selection of variables in our response to question 15.

Reviewer Comment 3.4 —

- It is mentioned in future work, but I think that it might be important at this stage of research to include conflict as a variable in the model.

Reply: Thanks a lot for the suggestion. We decided to complete the database and run experiments including variable characterizing conflicts, see Table 2. We included the annual sum of fatalities resulting

from conflict events over the polygon of interest from the ACLED database. Results show it as a relevant covariate that improves results and is a key driver of NDPs.

Reviewer Comment 3.5 — Minor comments

Introduction

1. In the second sentence, suggest removing “one of the main” and instead say that migration is one possible adaptation strategy
2. Can you clarify what you mean by “makes it difficult to implement effective measures”? Effective measures for what?
3. You should introduce what you mean by vulnerable/vulnerability before you claim “All of this aggravates the burden on the most vulnerable areas.” Also what burden? I think more clarity/ specificity would help.
4. Remove “certainly” in second paragraph
5. Why are environmental perceptions important? (Introduced in second paragraph)
6. What kind of insecurity are you referring to at the end of the second paragraph?
7. More generally- You might introduce the concept of forced displacement sooner/ restructure the introduction so that it is clearer. You mention it in the first and second paragraphs before really introducing what it means as a concept
8. Can you clarify what “three dimensions of the problem” you mean in the third paragraph?
9. Hazards, exposure, and vulnerability are three very specific, distinct concepts that deserve being introduced up front
10. Can you cite the sentence about how adaptive capacity and economic resources in a community matter for displacement?
11. Is it true that the majority of research focuses on international migration? How do you know? If it’s not known, then perhaps just say that much research focuses on international migration
12. Can you site the sentence about data limitations?
13. Not sure what you mean by “factorial complex problem”
14. Can you very briefly state what the sufficiency assumption is?
15. Why did you pick these variables?
16. A brief explanation/ justification of why you selected RF rather than another algorithm is warranted.

17. Can you more clearly motivate why you limit the analysis to low and middle income countries (also as defined by who)? I think introducing the concept of vulnerability more clearly in the intro, as mentioned, might help this
18. Is your time scale a year for the data?
19. Why did you select the thresholds that you did for Figure 5? (600 USD and 10m precipitation?)
20. You mention this as a next step, but I think it would be important to include conflict as a predictor for this work. It would be very interesting to know how strong of a predictor conflict is/ how your model performance improves.
21. I suggest citing this source as an example of using RF to study environmental migration- Best KB, Gilligan JM, Baroud H, Carrico AR, Donato KM, Ackerly BA, Mallick B. Random forest analysis of two household surveys can identify important predictors of migration in Bangladesh. *Journal of Computational Social Science*, 2020.

Reply: An item-by-item reply to each comment below:

1. Done.
2. Here we refer to measures that can prevent displacement, which include mitigation and adaptation strategies. These strategies can be used to either reduce hazards directly (e.g., building dikes to prevent flooding or reducing global emissions to slow climate change and extreme weather events) or to improve economic conditions in vulnerable areas, increasing the coping capacity and resources available for rebuilding and income diversification following damages to cultivated land, for instance.
3. We changed the sentence and made it more clear, avoiding the usage of specific terms which are introduced right after.
4. Done.
5. We mention the relevance of how people perceive their environment since, in the majority of cases, the decision of leaving involves also sentimental and psychological factors. Our work did not consider this dimension of the problem, since it is extremely hard to quantify, measure and model. Nonetheless, we thought it was important to mention it and cite some of the works that explore this complex sociological aspect of the displacement phenomenon.
6. We refer to all those conditions that challenge basic living standards, among which armed conflict, food and water scarcity. We clarified it in the main text.
7. Done, we now introduce the notion of displacement in the first paragraph.
8. The three dimensions would be hazard, exposure and vulnerability.
9. Done. We now introduce them early in the introduction, i.e. in the second paragraph.

10. We added Refs. 42-44. Communities facing displacement require adaptive capacity and economic resources to prepare for, respond to, and recover from it. Economic resources enable access to services and support, such as housing, employment, and education. Access to these resources helps reduce the impact of displacement and supports a successful transition into a new environment.
11. Done, we changed it in the text.
12. Yes, we added Ref. 60.
13. We rephrased it to “multivariate complex problem” for clarification in the new version of the manuscript. We were referring to the term “factorial” in the context of multivariate data analysis, where factorial refers to the analysis that examines the relationships between multiple variables thus allowing researchers to isolate the effects of one variable on another while controlling for all other variables. In layman’s terms, it is a method used to determine which variables are most important in producing certain results.
14. The sufficiency assumption in causal inference is a principle that states that all possible causes of a particular outcome have been included in the analysis. In other words, it assumes that if all possible causes are accounted for, the relationship between them and the outcome can be accurately assessed. In practice, any analysis must include all known causes of the outcome to be trustworthy. This is important because if any causes are included, the analysis could be accurate and complete. Of course, this can be fully achieved only in idealised contexts or in very controlled situations. In most observational studies one can assume that all (or most of) the relevant causes have been taken into account and perform checks a posteriori.
15. This study is the result of a two-year collaboration with IDMC. We aimed to characterize hazard-induced displacement by predicting the number of NDP using a small set of relevant covariates. To select the most useful covariates, we worked with IDMC experts and consulted previous literature. We then trained multiple models with potentially useful covariates to identify the most explanatory and non-collinear ones. Some more details are available in the Methods and Supplementary Information.
16. Following the suggestions from the other reviewers, we now added also a linear regression model, GBM models and the results of causal forest. Similar conclusions hold across different models (except for the simple linear regression which under-fits the data), thereby giving further support to our findings. Previously, we had also trained neural networks, Gaussian processes, and Poisson regression models, but RFs outperformed the rest. With such complex nonlinear relations, RFs perform particularly well where GPs take more work to train and reach lower performance. Besides, when compared to neural networks, RFs have been previously reported to work better when working with tabular data (see, e.g. <https://arxiv.org/abs/2207.08815>).
17. We included a vulnerability definition in the introduction and chose to focus on low and middle income countries for several reasons. Firstly, these countries are the most affected and vulnerable to weather extremes. Secondly, we believed that studying countries with comparable vulnerabilities would help interpret the results. Thirdly, disaster management policies are often absent or limited in poorer regions, while are overall more advanced in European and US countries that can count on a greater amount of resources. Finally, we used the recently introduced Relative Wealth Index (RWI), which is only available for low and middle income countries. In practice, we defined these

countries based on the availability of AWI/RWI, as stated in the paper and motivated extensively in Ref. 72.

18. As mentioned in the Methods section on page 7, given the starting date of the event, we consider a time window of 20 days for aggregating the predictors, which corresponds to the average duration of the recorded disasters. The NDPs are the total number of displaced persons registered by the IDMC for each event.
19. As also said in the paper, the blue and red curves for different thresholds have been added for illustrative purposes and we decided to change some of them in the revised version of the manuscript. The scatter plots of the Shapley values themselves provide all the necessary information, and the smoothed average lines were added for reader convenience.
20. Thanks a lot for the suggestion. We decided to complete the database and run experiments including variable characterizing conflicts. Results show it as a relevant covariate that improves results and is a key driver of NDPs.
21. We added the Ref. 52.

Reviewers' Comments:

Reviewer #2:

Remarks to the Author:

The authors have made a commendable effort to address reviewers' concerns and enhance the paper's methodology, and overall I think the paper has been strengthened as a result. Now that these more substantive issues have been addressed, I would suggest that the authors focus on a few other improvements, namely (1) simplifying/reframing the findings to make the paper easier for readers to digest, (2) Ensuring that results are interpreted with caution and that all statements are justified, (3) Checking consistency of figures and tables, (4) A thorough round of copy editing.

1. SIMPLIFYING/REFRAMING THE PAPER:

On this reading I feel that the paper is still somewhat complex, and I wonder if some basic reframing can help clarify the paper's message a bit. Specifically:

- The variables used are described in different ways: sometimes they are referred to environmental/societal/economic, sometimes economic/weather/land specific, sometimes hazard/exposure/vulnerability variables. I think I'd try to stick consistently to the latter framing, and in particular, map each variable to either hazard, exposure, or vulnerability early on in the paper. This is done on Page 27 but I think it would fit well when variables are introduced on Page 3. I think this is in line with R3's suggestion to more centrally emphasize these three factors.
- Once you've made this mapping more central, then I think you can more clearly draw out your findings using this language. As I see it, they are: (1) Worse hazard = worse displacement (i.e., precipitation is important); this conclusion seems obvious to me, but what is interesting is that this comes out more strongly from the ML methods than from the linear regression. (2) Worse exposure/vulnerability = worse displacement (again, expected, but good to confirm). (3) Differential vulnerability – interactions between factors – this seems like the most interesting thing that emerges from the analysis, so I think the paper can continue to stress this as the benefit of using this XAI approach.
- I think all of the core content needed is in the paper, but some basic reworking – like enforcing strong topic sentences, reiterating conclusions, smoothing transitions, etc. – could really enhance readability. As a simple example, the discussion section seems to cover three main points – a summary of work done and findings relative to the literature, a mention of data limitations, and a discussion of the benefits of this approach – but it is contained entirely in a single paragraph, making it hard to follow this arc.

2. INTERPRETING RESULTS WITH CAUTION

- The paper states, “*Correctly, all models predict a higher NDP when the affected area is larger... Conclusions for the population covariate are similar (see e.g. Fig. 9C), although it is classified as less important since it only considers human exposure.*” Since we are considering only human displacement, it was not immediately obvious to me that population should be less important, until the paper later explained that affected area could proxy other damage like infrastructure and crops which could lead to more severe displacement – I'd clarify the logic here. Furthermore, I would still avoid definitively saying that this is WHY population is classified as less important – I don't think we have enough information in the data/models to say why this is the case, so I'd use more

conservative language (e.g., “This is likely due to the fact that it only considers human exposure, whereas affected area might also capture infrastructure and crop damage which could worsen the severity of displacement.”)

- The paper states, “*LR downplays the importance of precipitation and assigns greater importance to wind speed. In contrast, the Shapley precipitation values for both GBM and RF models show that it is one of the two most important predictors of NDP. This can be explained by the highly non-linear relationship between NDP and precipitation, which emphasizes the need for ML models.*” I think it might be worth adding that in addition to the non-linear relationship between NDP and precipitation, the linear model will also fail to capture *interactions* between precipitation and other vulnerability/exposure factors, which the paper has shown are important.
- The paper considers that education and income are indicators of vulnerability. However, I imagine they are also highly correlated with other important factors – like the quality of governance and the effectiveness of a country’s disaster response. It might be worth just noting, as a limitation, these types of omitted confounding factors.
- I am encouraged by the fact that the causal forest results seem to match the XAI findings. However, the causal forest approach has found no variable with a significant causal effect (Table 3) – the paper should at least address/acknowledge this.
- The paper mentions that disasters with incomplete records were dropped, including those from most higher-income countries; at some point I might mention this as a limitation and discuss the potential bias this could introduce.
- You say “*storms and landslides are the hazards that cause greater damage to cultivated fields*” relative to floods – is this true? It seems that floods would damage cultivated fields, whereas landslides would occur in hilly areas that might not be cultivated?
- You say, “*We can see how, in presence of similar precipitation levels, NDP are greater for events in countries that depend more on agriculture. This is because if peoples’ livelihood depends on agriculture, then damages to the cultivated fields produced by weather-related disasters might result in forced displacements more often than in urban or industrialized countries, which are more resilient.*” – I might use more cautious language than stating “this is because”, since we know from the correlation table that agricultural land is also a proxy for wealth and I am not sure we can definitively disentangle the two; agricultural land is also positively correlated with conflict fatalities.

3. FIGURES/TABLES:

- I recommend going through all figures and (1) Ensuring axes are labeled (e.g. Figure 16); (2) Ensuring labels/annotations are precise (e.g. for confidence intervals, is this the 95% confidence interval?); (3) Ensuring that labels/annotations are consistent (e.g. figure 16 refers to “North America” and “N.C. America” and the order of continents changes between the top and bottom panels).
- In Table 2, (1) what is meant by “max,max” and “mean,mean” (why are these repeated?) (2) For AWI, why take max rather than mean or population-weighted values?
- Tables 5-7: I’d round these to integer values and add commas for ease of reading large numbers.

- Table 16, would it be relatively easy to also add a column with the number of events by country, so we can better understand which countries are best represented in the dataset?
- Figure 2: Remove latitude/longitude axes and labels
- Figure 5, what does “Average Causal Effect” refer to? Wouldn’t the boxplots be showing median causal effects? You say that this is the same information shown in Table 3, but I don’t think this is true.
- Figure 14: What are “country label” and “year-month label”? Are these numbers meaningful? I am guessing “country label” is just an ID, in this case maybe the histogram should be sorted by bar height since the order of labels is not meaningful? For “year-month label”, can we put the actual year-month labels? Why is there a break in the middle of the data where few or no disasters occurred?
- Figure 18: The text “1e11” randomly appears
- Figure 20: Is this figure necessary? What is gained by adding it?
- Figure 21: I wonder if it would be helpful to add a line of best fit to the scatter plots, for ease of interpretation of the general relationship?
- All included figures should be referenced at some point in the text.
- Tables are double-labeled, e.g. “Table 4” caption is followed by the text “Table 4”

4. CLARIFICATIONS/COPY EDITING:

- Each time you mention fatalities, can you specify “conflict” fatalities? My natural instinct is to assume that you are talking about disaster fatalities (which would be a proxy for severity), I kept having to stop and remind myself that this was not the case.
- P1: “although the widespread quest” → “despite the widespread quest”
- P2: “it is worth noticing that, many” → no comma
- P3: “In this way, we focus only on low and middle-income countries, “. → “Due to data limitations, we focus only on low and middle-income countries” (current wording is a bit vague/confusing)
- P3: One of the reviewers asked for clarification of the sufficiency assumption, which you included in the rebuttal, but can you also put this in the paper text?
- P4: You say, “feature importance for the LR (A) is calculated simply by multiplying the weights by the corresponding predictor values for each instance.” – Do you mean feature “contributions”? (I think this is what Shapely values are getting at). If the variables are normalized, I think the simplest measure of feature importance would just be the coefficients directly.
- P4: “Higher AWI leads to lower NDP” → as noted in my previous review, I would try to steer clear of this type of causal language and say “is associated with”
- P4: “Weather factors ,” has an extra space
- P5: “stressors that amplify stress conditions” = redundant
- P6: What is meant by “solid assumptions limit mechanistic models”? I understand that mechanistic models will not work here but I don’t quite understand what is being implied about assumptions.
- P6: “For what regards” → “In regard to”

- P6: “Extensions” → “extents”
- P7: “Forthcoming future” = redundant
- P7: “OpenStreetMap” is singular
- P7: “AWI data is available for most” → “AWI data is mostly available for” (or similar wording – the current wording does not make it clear that high-income countries are excluded)
- P8: Single quotation marks are not correctly typeset so they angle in the wrong direction
- P8: “More information on causal forestS”
- P16: “do a split” → “create a split”
- P18: “overall importance of each predictor” → would it be more accurate to say “overall contribution”?
- P20: q is not defined
- P23: “THE total number of NDP and average NDP”
- P29: “between all covariates between them” = redundant
- There should be a period before footnote #103

Reviewer #3:

Remarks to the Author:

I think that the authors have sufficiently responded to the reviewer comments.

Response to the reviewers: Paper # NCOMMS-22-40518

Reviewer Comment 0.1 — The authors have made a commendable effort to address reviewers' concerns and enhance the paper's methodology, and overall I think the paper has been strengthened as a result. Now that these more substantive issues have been addressed, I would suggest that the authors focus on a few other improvements, namely (1) simplifying/reframing the findings to make the paper easier for readers to digest, (2) Ensuring that results are interpreted with caution and that all statements are justified, (3) Checking consistency of figures and tables, (4) A thorough round of copy Editing.

Reply: Thanks for the positive evaluation of our work, and the constructive comments to improve it, as well as the thorough revision of style and formatting issues. In the following, we answer all reviewer's comments point-by-point, and implement the changes in the new version of the manuscript highlighted in blue.

Reviewer Comment 0.2 — The variables used are described in different ways: sometimes they are referred to environmental/societal/economic, sometimes economic/weather/land specific, sometimes hazard/exposure/vulnerability variables. I think I'd try to stick consistently to the latter framing, and in particular, map each variable to either hazard, exposure, or vulnerability early on in the paper. This is done on Page 27 but I think it would fit well when variables are introduced on Page 3. I think this is in line with R3's suggestion to more centrally emphasize these three factors.

Reply: We've standardized how we refer to the different variables in the study to enhance consistency and clarity, as recommended. On page 3, when introducing the predictor variables, we explicitly link them to the hazard-exposure-vulnerability triangle. However, it's important to emphasize that this linkage isn't always straightforward. In the field of disaster risk literature, it's well recognized that some of these dimensions interact with each other. Additionally, in certain cases, the same variables may serve as proxies for both exposure and vulnerability components, potentially leading to some degree of misinterpretation if strictly categorized into these three groups. We've addressed this limitation by including a couple of sentences discussing it.

Reviewer Comment 0.3 — Once you've made this mapping more central, then I think you can more clearly draw out your findings using this language. As I see it, they are: (1) Worse hazard = worse displacement (i.e., precipitation is important); this conclusion seems obvious to me, but what is interesting is that this comes out more strongly from the ML methods than from the linear regression. (2) Worse exposure/vulnerability = worse displacement (again, expected, but good to confirm). (3) Differential vulnerability – interactions between factors – this seems like the most interesting thing that emerges from the analysis, so I think the paper can continue to stress this as the benefit of using this XAI approach.

Reply: We appreciate the Reviewer for highlighting the key achievements of the approach we employ in the paper. This involves identifying evidence of varying vulnerability through a purely data-driven method, without relying on strong assumptions, except for the utilization of a broad range of potentially explanatory variables. In response to the Reviewer's suggestion, we have incorporated additional comments in both the results and conclusions sections to emphasize the advantages of employing this approach. Please refer to the sections highlighted in blue in the revised version of the manuscript.

Reviewer Comment 0.4 — I think all of the core content needed is in the paper, but some basic reworking – like enforcing strong topic sentences, reiterating conclusions, smoothing transitions, etc. – could really enhance readability. As a simple example, the discussion section seems to cover three main points – a summary of work done and findings relative to the literature, a mention of data limitations, and a discussion of the benefits of this approach – but it is contained entirely in a single paragraph, making it hard to follow this arc.

Reply: We have made an extra effort to improve the readability of the paper by simplifying messages, removing redundancies and restructuring conclusions. All these changes have been highlighted in blue in the new version of the manuscript.

Reviewer Comment 0.5 — The paper states, “Correctly, all models predict a higher NDP when the affected area is larger. . . Conclusions for the population covariate are similar (see e.g. Fig. 9C), although it is classified as less important since it only considers human exposure.” Since we are considering only human displacement, it was not immediately obvious to me that population should be less important, until the paper later explained that affected area could proxy other damage like infrastructure and crops which could lead to more severe displacement – I’d clarify the logic here. Furthermore, I would still avoid definitively saying that this is why population is classified as less important – I don’t think we have enough information in the data/models to say why this is the case, so I’d use more conservative language (e.g., “This is likely due to the fact that it only considers human exposure, whereas affected area might also capture infrastructure and crop damage which could worsen the severity of displacement.”)

Reply: We have now clarified early in the text that the variable “area” acts as a stand-in for both human and infrastructure (as well as facilities, buildings, and so on) exposures, while “population” exclusively represents human exposure. Although establishing a causal relationship is a complex task, we believe we have reasonably addressed the sufficiency assumption and considered many potential confounding factors, see also answers below. At the same time, we acknowledge that limitations in available data and unobserved variables could potentially lead to misguided or weak conclusions. In line with the Reviewer’ suggestion, we have rephrased the sentence at the top of page 6 to adopt a more cautious tone, refraining from making definitive causal claims.

Reviewer Comment 0.6 — The paper states, “LR downplays the importance of precipitation and assigns greater importance to wind speed. In contrast, the Shapley precipitation values for both GBM and RF models show that it is one of the two most important predictors of NDP. This can be explained by the highly non-linear relationship between NDP and precipitation, which emphasizes the need for ML models.” I think it might be worth adding that in addition to the non-linear relationship between NDP and precipitation, the linear model will also fail to capture *interactions* between precipitation and other vulnerability/exposure factors, which the paper has shown are important.

Reply: We added a comment about interacting terms, as suggested.

Reviewer Comment 0.7 — The paper considers that education and income are indicators of vulnerability. However, I imagine they are also highly correlated with other important factors – like the quality of governance and the effectiveness of a country’s disaster response. It might be worth just noting, as a limitation, these types of omitted confounding factors.

Reply: We agree with the Reviewer and are grateful for highlighting this point. We added a comment on the sufficiency assumption and the limitations of the study in the “Building a Global Dataset of Displacements” subsection of the Methods section. See the added paragraph on page 8. We also added a sentence in the second paragraph on page 6, where we discuss the results for education expenses.

Reviewer Comment 0.8 — I am encouraged by the fact that the causal forest results seem to match the XAI findings. However, the causal forest approach has found no variable with a significant causal effect (Table 3) – the paper should at least address/acknowledge this.

Reply: Causal forests primarily reveal variations in a causal effect but do not establish causality themselves. A standard causal forest assumes that the assignment to treatment is exogenous, which places some constraints on the true causal mechanisms inferred from it. Although there are extensions of causal forests that allow for covariate adjustment or instrumental variables, we did not utilize them in our study. Our main aim was to evaluate whether the identified relationships (and thus our conclusions) could be interpreted causally. As more data, particularly time series data, become available, it would be worthwhile to further investigate the existence of causal mechanisms using advanced causal inference techniques, cause-effect estimation methods, and causal discovery algorithms. We have added a comment to acknowledge this potential limitation.

Reviewer Comment 0.9 — The paper mentions that disasters with incomplete records were dropped, including those from most higher-income countries; at some point I might mention this as a limitation and discuss the potential bias this could introduce.

Reply: We acknowledge that such limitations should be further stressed in the paper. We added some additional comments on the sufficiency assumption and the limitations of the study in the “Building a Global Dataset of Displacements” subsection of the Methods section. See also the paragraph highlighted in blue on page 8.

Reviewer Comment 0.10 — You say “storms and landslides are the hazards that cause greater damage to cultivated fields” relative to floods – is this true? It seems that floods would damage cultivated fields, whereas landslides would occur in hilly areas that might not be cultivated?

Reply: Storms and landslides were observed to cause slightly more damage to cultivated fields compared to floods. The impact of these hazards can vary widely depending on the geographical context and local conditions. Landslides can affect crop-fields both directly and indirectly, such as by burying crops under debris, eroding topsoil, changing drainage patterns, impeding access, inducing slope instability, and posing risks to livestock. In summary, as we stress in the main text, all three types of hazards can inflict significant damage on agriculture. The subtle differences among them, as highlighted in our manuscript, may be influenced by various other factors specific to the location in question and the sample of events under consideration.

Reviewer Comment 0.11 — You say, “We can see how, in presence of similar precipitation levels, NDP are greater for events in countries that depend more on agriculture. This is because if peoples’ livelihood depends on agriculture, then damages to the cultivated fields produced by weather-related disasters might result in forced displacements more often than in urban or industrialized countries, which are more resilient.” – I might use more cautious language than stating “this is because”,

since we know from the correlation table that agricultural land is also a proxy for wealth and I am not sure we can definitively disentangle the two; agricultural land is also positively correlated with conflict fatalities.

Reply: We acknowledge the need for caution in our language, and rephrased that statement as follows: "We observe that, in the presence of similar precipitation levels, NDP tends to be higher in countries with a greater dependence on agriculture. This association may be linked to the vulnerability of livelihoods tied to agriculture, potentially leading to more frequent forced displacements than in urban or industrialized countries, although we recognize that agricultural dependence can also correlate with other factors, such as wealth and conflict."

Reviewer Comment 0.12 — I recommend going through all figures and (1) Ensuring axes are labeled (e.g. Figure 16); (2) Ensuring labels/annotations are precise (e.g. for confidence intervals, is this the 95% confidence interval?); (3) Ensuring that labels/annotations are consistent (e.g. figure 16 refers to "North America" and "N.C. America" and the order of continents changes between the top and bottom panels). In Table 2, (1) what is meant by "max,max" and "mean,mean" (why are these repeated?) (2) For AWI, why take max rather than mean or population-weighted values?

Reply: Those refer to the spatial and temporal aggregation and it was indeed not clear in the table. We have modified Table 2 with a temporal and spatial aggregation column. We added labels to the axes. Initially we tested AWI_{mean} , AWI_{max} and AWI_{median} as variables in the models. Finally, we opted for AWI_{max} because in AWI calculations, non-populated areas display very low AWI values, whereas the presence of buildings and populations can significantly elevate its value. This choice allows us to more effectively pinpoint the presence of human settlements for a given polygon. We did not take population-weighted values because *Population* is already a variable in the model whose interactions with AWI should be captured.

Reviewer Comment 0.13 — Tables 5-7: I'd round these to integer values and add commas for ease of reading large numbers.

Reply: Done

Reviewer Comment 0.14 — Table 16, would it be relatively easy to also add a column with the number of events by country, so we can better understand which countries are best represented in the dataset?

Reply: Yes, we added a column with the number of events per country.

Reviewer Comment 0.15 — Figure 2: Remove latitude/longitude axes and labels.

Reply: Done

Reviewer Comment 0.16 — Figure 5, what does "Average Causal Effect" refer to? Wouldn't the boxplots be showing median causal effects? You say that this is the same information shown in Table 3, but I don't think this is true.

Reply: Yes, they display the median and quartiles, from which the mean can be easily deduced. Typically, this procedure is referred to as Average Treatment Effect or Average Causal Effect, and we

maintain this notation in our paper. Nevertheless, we have updated the caption under Figure 5 and further explained these nuances in the text.

Reviewer Comment 0.17 — Figure 14: What are “country label” and “year-month label”? Are these numbers meaningful? I am guessing “country label” is just an ID, in this case maybe the histogram should be sorted by bar height since the order of labels is not meaningful? For “yearmonthlabel”, can we put the actual year-month labels? Why is there a break in the middle of the data where few or no disasters occurred? Figure 18: The text “1e11” randomly appears

Reply: We appreciate the Reviewer’s questions. To conduct spatial and temporal cross-validation, we assign a country ID or label to each event for the former, and for the latter, we use a label or ID based on the combination of both the year and month. As per the Reviewer’s request, we have adjusted the label names to enhance clarity. Additionally, we have made corrections to Figure 18.

Reviewer Comment 0.18 — Figure 20: Is this figure necessary? What is gained by adding it?

Reply: We agree with the Reviewer on the fact that this Figure in the supplementary materials is not necessary and, thus, we removed it.

Reviewer Comment 0.19 — Figure 21: I wonder if it would be helpful to add a line of best fit to the scatter plots, for ease of interpretation of the general relationship?

Reply: All the relationships depicted exhibit a high degree of non-linearity. Given this, we hold the belief that incorporating such lines would not substantially enhance the visualization of meaningful trends, and in fact, they might potentially lead to confusion for the reader. For this reason we decided to display only the raw data points in the form of scatter plots without fitting any (univariate) linear model which would necessarily have very low predictive performance.

Reviewer Comment 0.20 — All included figures should be referenced at some point in the text

Reply: Done.

Reviewer Comment 0.21 — Tables are double-labeled, e.g. “Table 4” caption is followed by the text “Table 4”

Reply: Corrected.

Reviewer Comment 0.22 — Each time you mention fatalities, can you specify “conflict” fatalities? My natural instinct is to assume that you are talking about disaster fatalities (which would be a proxy for severity), I kept having to stop and remind myself that this was not the case.

Reply: We changed in the text “fatalities” for “conflict fatalities”.

Reviewer Comment 0.23 — P1: “although the widespread quest” à “despite the widespread quest”

P2: “it is worth noticing that, many” à no comma

P3: “In this way, we focus only on low and middle-income countries, “. à “Due to data limitations, we focus only on low and middle-income countries” (current wording is a bit vague/confusing)

Reply: All corrected.

Reviewer Comment 0.24 — P3: One of the reviewers asked for clarification of the sufficiency assumption, which you included in the rebuttal, but can you also put this in the paper text?

Reply: We included the clarification in the main text at the beginning of page 3 where we mention the sufficiency assumption in causal inference.

Reviewer Comment 0.25 — P4: You say, “feature importance for the LR (A) is calculated simply by multiplying the weights by the corresponding predictor values for each instance.” – Do you mean feature “contributions”? (I think this is what Shapely values are getting at). If the variables are normalized, I think the simplest measure of feature importance would just be the coefficients directly.

Reply:

Thank you for your question. While it’s true that for LR we could examine the coefficients directly, we chose to present the product of the coefficients and the feature values. This allows for a more direct comparison of the LR results with the Shapley values obtained for RF and GBM. In cases of model linearity, the Shapley values essentially simplify to the values of the dependent variable multiplied by their corresponding coefficients. This approach helps to maintain consistency across our analyses.

Reviewer Comment 0.26 — P4: “Higher AWI leads to lower NDP” à as noted in my previous review, I would try to steer clear of this type of causal language and say “is associated with”

P4: “Weather factors ,” has an extra space

P5: “stressors that amplify stress conditions” = redundant

Reply: All corrected.

Reviewer Comment 0.27 — P6: What is meant by “solid assumptions limit mechanistic models”? I understand that mechanistic models will not work here but I don’t quite understand what is being implied about assumptions.

Reply: Typical assumptions involve: the linearity of the relation between target and covariates, or assuming an explicit functional (or distributional) form of interactions when they are included. In contrast, none of these assumptions are needed when using RF or GBM. We clarified the sentence in the manuscript.

Reviewer Comment 0.28 — P6: “For what regards” à “In regard to”

P6: “Extensions” à “extents”

P7: “Forthcoming future” = redundant

P7: “OpenStreetMap” is singular

P7: “AWI data is available for most” à “AWI data is mostly available for” (or similar wording – the current wording does not make it clear that high-income countries are excluded)

P8: Single quotation marks are not correctly typeset so they angle in the wrong direction

P8: “More information on causal forests”

P16: “do a split” à “create a split”

P18: “overall importance of each predictor” à would it be more accurate to say “overall contribution”?

P20: q is not defined

P23: “THE total number of NDP and average NDP”

P29: “between all covariates between them” = redundant

Reply: All corrected.